# Stop-Think-AutoRegress: Language Modeling with Latent Diffusion Planning

**Justin Lovelace,**\* **Christian Belardi, Sofian Zalouk, Adhitya Polavaram,**
**Srivatsa Kundurthy, Kilian Q Weinberger**
Department of Computer Science, Cornell University

## Abstract

The Stop-Think-AutoRegress Language Diffusion Model (STAR-LDM) integrates latent diffusion planning with autoregressive generation. Unlike conventional autoregressive language models limited to token-by-token decisions, STAR-LDM incorporates a "thinking" phase that pauses generation to refine a semantic plan through diffusion before continuing. This enables global planning in continuous space prior to committing to discrete tokens. Evaluations show STAR-LDM significantly outperforms similar-sized models on language understanding benchmarks and achieves $> 70\%$ win rates in LLM-as-judge comparisons for narrative coherence and commonsense reasoning. The architecture also allows straightforward control through lightweight classifiers, enabling fine-grained steering of attributes without model retraining while maintaining better fluency-control trade-offs than specialized approaches. Our code is available at https://github.com/justinlovelace/STAR-LDM.

## 1 Introduction

Large Language Models (LLMs), typically based on the Transformer architecture (Vaswani et al., 2017), have demonstrated remarkable capabilities across a vast range of natural language tasks (Kojima et al., 2022; Brown et al., 2020). Their success largely stems from the autoregressive (AR) generation paradigm, where text is produced sequentially, predicting one token at a time conditioned on the preceding context. However, this strictly left-to-right, token-by-token generation process fundamentally differs from human writing practices. Human authors frequently pause, reflect, plan ahead, and revise their text to ensure global coherence, stylistic consistency, and adherence to overarching goals Hayes & Flower (1996). In contrast, standard autoregressive models make irrevocable, local decisions at each step, which inherently limits their capacity for long-range planning or dynamic control over the generation process (Liu et al., 2021b). Consequently, ensuring that generated text globally satisfies complex constraints, maintains a consistent persona, or avoids undesirable content purely through local token predictions remains a significant challenge, often necessitating computationally expensive fine-tuning or reinforcement learning procedures (Ziegler et al., 2019; Ouyang et al., 2022).

Recent advancements in generative modeling, particularly score-based diffusion models (Ho et al., 2020; Song et al., 2020), offer a fundamentally different paradigm. These models generate data through an iterative denoising process, beginning with pure noise and guided by a learned score function. Crucially, this score function can be readily modified during generation to incorporate external guidance signals from lightweight classifiers (Dhariwal & Nichol, 2021; Ho & Salimans, 2021), enabling flexible control over various attributes of the generated output (e.g., style, content). This inherent controllability, which can be effectively applied in continuous spaces such as image latents (Rombach et al., 2021) or text embeddings (Li et al., 2022; Lovelace et al., 2023), presents a particularly promising direction for enhancing control mechanisms in large language models.

---

\* Corresponding author: jl3353@cornell.edu

Inspired by the planning capabilities suggested by human writing and the controllability of diffusion models, we propose *Stop-Think-AutoRegress Language Diffusion Model (STAR-LDM)*. STAR-LDM is a novel, unified architecture that integrates latent diffusion planning directly into the autoregressive generation framework. It introduces a "thinking" phase where standard token generation is paused. During this phase, the model utilizes a diffusion process operating in a continuous sentence embedding space to refine a latent representation (a soft prompt) that encodes a plan for the subsequent text. This refined latent representation then guides the resumption of autoregressive token generation. By jointly training the autoregressive and diffusion components, STAR-LDM learns to perform global semantic planning in a continuous space *before* committing to discrete token choices, enabling more coherent and controllable generation.

Our contributions are as follows: (1) We propose STAR-LDM, a unified architecture that integrates latent diffusion planning within an autoregressive language model. (2) We demonstrate that STAR-LDM achieves improved language generation quality. (3) We show that our unified architecture leads to substantial improvements on standard language understanding benchmarks, indicating that the planning mechanism enhances, rather than hinders, core language capabilities. (4) We validate that STAR-LDM's integrated diffusion component enables efficient, plug-and-play control over text attributes (like sentiment and toxicity) using lightweight classifiers during inference, without requiring model retraining. STAR-LDM represents a step towards language models that can "stop and think" entirely in latent space, leading to higher-quality, more controllable, and potentially more reliable text generation.

## 2 Background

Diffusion models (Sohl-Dickstein et al., 2015; Ho et al., 2020; Song et al., 2020; Kingma & Gao, 2023), are a class of generative models that iteratively refine a sample of noise to a sample from some data distribution in a coarse-to-fine manner. Given clean data $\mathbf{x}_{\text{data}}$ from $q(\mathbf{x}_{\text{data}})$, diffusion models learn a model $p_\theta(\mathbf{x}_{\text{data}})$ approximating $q(\mathbf{x}_{\text{data}})$. Here, $\mathbf{x}_{\text{data}}$ could be images, audio, or continuous latent representations relevant to NLP (Li et al., 2022; Rombach et al., 2021; Lovelace et al., 2024), such as the sentence embeddings used in this work (Ni et al., 2022).

**Forward process.** Diffusion models define a forward noising process that transitions clean data $\mathbf{x}_{\text{data}}$ to noise over continuous time $t \in [0, 1]$. The noisy latent $\mathbf{z}_t$ at time $t$ is given by $q(\mathbf{z}_t|\mathbf{x}_{\text{data}}) = \mathcal{N}(\mathbf{z}_t; \alpha_t \mathbf{x}_{\text{data}}, \sigma_t^2 \mathbf{I})$, which can be sampled as:

$$\mathbf{z}_t = \alpha_t \mathbf{x}_{\text{data}} + \sigma_t \boldsymbol{\epsilon}, \quad \text{where} \quad \boldsymbol{\epsilon} \sim \mathcal{N}(\mathbf{0}, \mathbf{I}). \tag{1}$$

Using the variance-preserving formulation (Ho et al., 2020) ($\sigma_t^2 = 1 - \alpha_t^2$), the noise level is parameterized by the log Signal-to-Noise Ratio (SNR), $\lambda_t = \log(\alpha_t^2/\sigma_t^2)$. The schedule $\alpha_t$ decreases such that $\mathbf{z}_0 \approx \mathbf{x}_{\text{data}}$ and the distribution of $\mathbf{z}_1$ approaches a standard normal, i.e., $q(\mathbf{z}_1|\mathbf{x}_{\text{data}}) \approx \mathcal{N}(\mathbf{0}, \mathbf{I})$.

**Generative model.** The generative process reverses the forward process, starting from noise $\mathbf{z}_1 \sim \mathcal{N}(\mathbf{0}, \mathbf{I})$ and iteratively denoising it to produce a final sample $\mathbf{z}_0$ that approximates the data distribution $q(\mathbf{x}_{\text{data}})$. This requires the score function $\nabla_{\mathbf{z}_t} \log p_t(\mathbf{z}_t)$, the gradient of the log-density of noisy data at time $t$. Diffusion models introduce a neural network $\mathbf{s}_\theta(\mathbf{z}_t, t)$ to approximate this score. Sampling involves iteratively applying the learned model $\mathbf{s}_\theta$ (or an equivalent parameterization, like noise prediction) to reverse the diffusion from $t = 1$ to $t = 0$, often using solvers like DDPM (Ho et al., 2020).

**Training objective.** The score network $\mathbf{s}_\theta$ can be learned via denoising score matching (Song & Ermon, 2019). An equivalent and common objective (Ho et al., 2020) trains a network $\boldsymbol{\epsilon}_\theta(\mathbf{z}_t, t)$ to predict the noise $\boldsymbol{\epsilon}$ added to $\mathbf{x}_{\text{data}}$ to get $\mathbf{z}_t$, using a loss like:

$$\mathcal{L}(\theta) = \mathbb{E}_{t, \mathbf{x}_{\text{data}} \sim q(\mathbf{x}_{\text{data}}), \boldsymbol{\epsilon} \sim \mathcal{N}(\mathbf{0}, \mathbf{I})} \left[ w(\lambda_t) \cdot \|\boldsymbol{\epsilon} - \boldsymbol{\epsilon}_\theta(\mathbf{z}_t, t)\|_2^2 \right], \tag{2}$$

where $\mathbf{z}_t = \alpha_t \mathbf{x}_{\text{data}} + \sigma_t \boldsymbol{\epsilon}$ and $w(\lambda_t)$ is an SNR-dependent weighting term (Kingma et al., 2021). The score and noise predictions are related by $\mathbf{s}_\theta(\mathbf{z}_t, t) = -\boldsymbol{\epsilon}_\theta(\mathbf{z}_t, t)/\sigma_t$.

**Plug-and-play control.** A key advantage of diffusion, as opposed to autoregressive, models is their controllability via plug-and-play guidance (Song et al., 2020; Dhariwal & Nichol, 2021). To generate samples conditioned on $\mathbf{y}$, the sampling process is guided by the conditional score $\nabla_{\mathbf{z}_t} \log p_t(\mathbf{z}_t|\mathbf{y})$. Using Bayes' rule, this conditional score can be decomposed:

$$\nabla_{\mathbf{z}_t} \log p_t(\mathbf{z}_t|\mathbf{y}) = \underbrace{\nabla_{\mathbf{z}_t} \log p_t(\mathbf{z}_t)}_{\text{Unconditional Score}} + \underbrace{\nabla_{\mathbf{z}_t} \log p_t(\mathbf{y}|\mathbf{z}_t)}_{\text{Guidance Term}}. \tag{3}$$

The first term is the standard score approximated by the unconditional diffusion model $\mathbf{s}_\theta(\mathbf{z}_t, t)$. The second term guides the sampling towards latents $\mathbf{z}_t$ that are likely to produce the condition $\mathbf{y}$. This guidance term can be approximated using gradients from a *separate classifier* trained on noisy data, which is often called *classifier guidance* (Dhariwal & Nichol, 2021).

In practice, a guidance scale $s \geq 0$ is introduced to modulate the strength of the condition:

$$\hat{\mathbf{s}}(\mathbf{z}_t, t, \mathbf{y}) = \nabla_{\mathbf{z}_t} \log p_t(\mathbf{z}_t) + s \cdot \nabla_{\mathbf{z}_t} \log p_t(\mathbf{y}|\mathbf{z}_t). \tag{4}$$

Setting $s = 0$ yields unconditional generation, while $s > 1$ increases the influence of $\mathbf{y}$. This provides flexible "plug-and-play" control without retraining the base model for each condition $\mathbf{y}$. This inherent controllability offers a promising avenue to address the control challenges of autoregressive language generation, which motivates our work.

## 3 Stop-Think-AutoRegress Language Diffusion Model

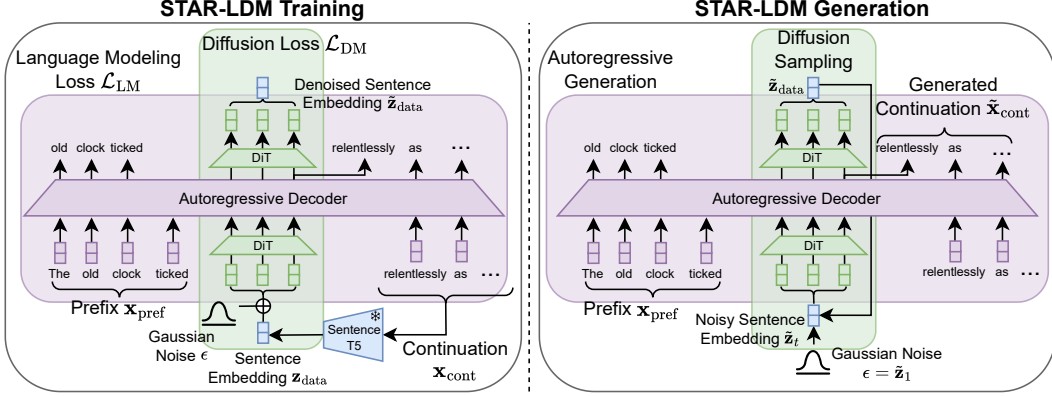

Figure 1: Unified diffusion-guided language model architecture illustrating the training (left) and generation (right) processes. See text for details.

We propose *Stop-Think-AutoRegress Language Diffusion Model (STAR-LDM)*, a unified architecture that augments autoregressive language modeling with latent diffusion planning. STAR-LDM can pause generation, plan in a continuous latent space, and resume autoregressive token generation with improved coherence.

Figure 1 depicts our proposed architecture, which consists of three major components, each with a primary role in enabling our "stop-think-autoregress" approach: (1) The **Autoregressive Decoder** (in purple) serves as the foundation of our model, handling token generation in a sequential manner similar to traditional language models. This component maintains the local fluency and grammatical structure necessary for coherent text. (2) The continuous **Sentence Embedding Space** (blue vectors) provides a latent planning space that captures semantic meaning. Using 768-dimensional Sentence-T5 XL embeddings (Ni et al., 2022), this space represents text at a higher level of abstraction than tokens, with representations that capture core semantic content while being invariant to shallow. (3) Two **Diffusion Transformers (DiTs)** (Peebles & Xie, 2022) (in green) facilitate the translation between the continuous embedding space and the discrete token representations used by the decoder. These transformer encoders are also conditioned on the noise level through adaptive normalization layers following Peebles & Xie (2022).

In our approach, text generation is conceptualized as a two-part process: first processing a *prefix* (the initial portion of text that has already been generated), then planning and generating a coherent *continuation*. The prefix might be a prompt provided by a user or previously generated text, while the continuation represents new text the model will generate with guidance from a semantic plan.

During training, these components work together as follows. First, the autoregressive decoder processes the prefix text $\mathbf{x}_{\text{pref}}$ ("The old clock ticked") like a standard autoregressive language model. The model also processes a noisy representation of the continuation's semantic meaning. This representation is created by encoding the ground truth continuation $\mathbf{x}_{\text{cont}}$ ("relentlessly as...") into a clean embedding $\mathbf{z}_{\text{data}}$ using Sentence-T5, then adding noise according to the diffusion schedule to obtain $\mathbf{z}_t = \alpha_t \mathbf{z}_{\text{data}} + \sigma_t \epsilon$.

We linearly project this noisy embedding into a sequence of 8 soft vectors, which the first DiT processes (conditioning on the current noise level) to form a soft prompt. This prompt will both be used to guide diffusion denoising and subsequent token generation. For diffusion denoising, the autoregressive decoder naturally incorporates information from the prefix as it processes this soft prompt. The decoder's outputs then flow to the second DiT, which generates the final representations for the diffusion prediction. We apply a learned linear projection to obtain the final diffusion prediction $\tilde{\mathbf{z}}_{\text{data}}$, completing a step of the denoising process.

After processing this soft prompt, the decoder reverts to standard next-token prediction. The predictions for the text continuation are therefore conditioned on both the textual prefix and the (noisy) sentence embedding of that text continuation. The sentence embedding provides a valuable signal for the token prediction objective, teaching the decoder to generate text aligned with the sentence embedding.

This integrated design creates a model that can simultaneously handle token-level prediction and semantic-level planning. As noise levels decrease during training or sampling, the semantic guidance becomes increasingly precise, enabling the model to generate continuations that follow the semantic content in the denoised embedding while maintaining both local fluency and global coherence.

### 3.1 Training Procedure

Our training approach combines autoregressive language modeling with diffusion-based planning in a multi-task learning framework. This training strategy will enable STAR-LDM to learn both token generation and semantic planning simultaneously. During self-supervised pre-training (left side of Figure 1), we sample text segments and randomly split each into a prefix $\mathbf{x}_{\text{pref}}$ and continuation $\mathbf{x}_{\text{cont}}$. This simulates stopping points where the model would pause to plan. We jointly optimize with two objectives:

**Language Modeling Loss ($\mathcal{L}_{\text{LM}}$).** The loss combines standard prefix modeling and noise-conditioned continuation modeling:

$$\mathcal{L}_{\text{LM}} = \underbrace{-\sum_{i=1}^{p} \log P(x_{\text{pref},i}|\mathbf{x}_{\text{pref},<i}; \theta)}_{\text{Prefix modeling}} \underbrace{-\sum_{j=1}^{n-p} \log P(x_{\text{cont},j}|\mathbf{x}_{\text{pref}}, \mathbf{x}_{\text{cont},<j}, \mathbf{z}_t; \theta)}_{\text{Continuation modeling}}, \tag{5}$$

where $p$ is the prefix length, $n$ the total length, and $\mathbf{z}_t$ the noisy embedding of $\mathbf{x}_{\text{cont}}$.

**Diffusion Loss ($\mathcal{L}_{\text{DM}}$).** We encode the continuation $\mathbf{x}_{\text{cont}}$ using Sentence-T5 to obtain the clean embedding $\mathbf{z}_0$. Following standard diffusion training, we then sample a timestep $t$ and noise vector $\epsilon \sim \mathcal{N}(\mathbf{0}, \mathbf{I})$, compute the noisy embedding $\mathbf{z}_t = \alpha_t \mathbf{z}_0 + \sigma_t \epsilon$, and train the model to predict the added noise conditioned on the prefix and timestep:

$$\mathcal{L}_{\text{DM}} = \mathbb{E}_{t, \mathbf{x}_{\text{cont}}, \epsilon} \left[ w(\lambda_t) \cdot \|\epsilon - \epsilon_\theta(\mathbf{z}_t, t, \mathbf{x}_{\text{pref}})\|_2^2 \right] \tag{6}$$

This objective is weighted by $w(\lambda_t)$, a function of the log(SNR) that balances the importance of different noise levels[1]. The final training objective combines both losses, $\mathcal{L} = \mathcal{L}_{\text{LM}} + \beta\mathcal{L}_{\text{DM}}$, where the scalar $\beta$ balances the two components. STAR-LDM is therefore trained for autoregression and diffusion in a multi-task manner.

**Noise Conditioning.** The diffusion training process exposes the autoregressive decoder to embeddings $\mathbf{z}_t$ with varying noise levels, from $\alpha_1 \approx 0$ (pure noise) to $\alpha_0 \approx 1$ (clean). Like prior work that has utilized noise-conditioning to improve robustness (Ho et al., 2022; Lovelace et al., 2024), our decoder learns to adapt based on noise level: relying heavily on the semantic plan at low noise while falling back to standard prefix conditioning at high noise. This creates a natural control parameter during inference—the noise level—that adjusts the balance between diffusion planning and autoregressive generation.

## 3.2 Generation Process

While our training process optimizes STAR-LDM to both generate coherent text and denoise semantic embeddings, the inference process leverages these capabilities in a sequential "Stop-Think-AutoRegress" workflow. During inference, we assume a prompted setting where the user provides a prefix, and the model plans and generates a continuation.

During inference (Figure 1, right), our approach unfolds in three distinct phases: 1) **Stop**: The model processes the input prompt $\mathbf{x}_{\text{pref}}$ (provided by the user) through the autoregressive decoder. 2) **Think**: The model then "thinks" by sampling initial noise $\tilde{\mathbf{z}}_1 = \epsilon \sim \mathcal{N}(0, \mathbf{I})$ and performing iterative denoising to obtain a semantic plan. This iterative denoising process, guided by the prefix $\mathbf{x}_{\text{pref}}$, gradually transforms random noise into a coherent semantic embedding, $\tilde{\mathbf{z}}_0$.

Starting from pure noise $\tilde{\mathbf{z}}_1$, the model applies its trained noise prediction network $\epsilon_\theta$, consisting of a pass through the network, over multiple timesteps (e.g. 50 steps). At each step $t$, the model uses the predicted noise component to reduce the noise level of the embedding following a predetermined schedule. This process iteratively refines a coherent plan consistent with the given prefix. The generation process concludes with the third phase: 3) AutoRegress. Here, the generated semantic plan $\tilde{\mathbf{z}}_0$ guides the autoregressive decoder to produce a fluent continuation $\tilde{\mathbf{x}}_{\text{cont}}$. This is achieved by processing $\tilde{\mathbf{z}}_0$ through the first DiT to create a soft prompt that conditions the decoder alongside $\mathbf{x}_{\text{pref}}$. Once conditioned on this combined prompt, the decoder generates the continuation via standard autoregressive sampling.

**Prompt guidance.** During the thinking phase, we can optionally strengthen the influence of the prefix on the generated plan using classifier-free guidance (Ho & Salimans, 2021). This technique combines an unconditional prediction (ignoring the prefix) with a conditional prediction (using the prefix) to amplify the prefix's influence:

$$\tilde{\epsilon}_t = \epsilon_\theta(\mathbf{z}_t, t, \varnothing) + w \cdot (\epsilon_\theta(\mathbf{z}_t, t, \mathbf{x}_{\text{pref}}) - \epsilon_\theta(\mathbf{z}_t, t, \varnothing)) \tag{7}$$

where $w$ is the guidance scale and $\epsilon_\theta(\mathbf{z}_t, t, \varnothing)$ represents an unconditional prediction obtained by masking the prefix embeddings from the diffusion model. Higher values of $w$ increase the prompt's influence on the semantic plan.

**Plug-and-Play Control with Classifier Guidance.** To enable efficient classifier guidance (Dhariwal & Nichol, 2021), we train an MLP that directly operates on noisy sentence embeddings:

$$p_t(\mathbf{y}|\mathbf{z}_t) \approx f_\phi(\mathbf{z}_t, t) = \text{MLP}(\mathbf{z}_t, \text{TimeEmb}(t)), \tag{8}$$

where $\text{TimeEmb}(t)$ encodes the noise level via sinusoidal embeddings. To make the MLP robust to varying noise levels, we follow Dhariwal & Nichol (2021) and train on labeled data that we perturb with randomly chosen noise levels. During inference, we compute the guidance gradient directly:

$$\nabla_{\mathbf{z}_t} \log p_t(\mathbf{y}|\mathbf{z}_t) \approx \nabla_{\mathbf{z}_t} f_\phi(\mathbf{z}_t, t) \tag{9}$$

---

[1]We adopt the sigmoid weighting and cosine noise schedule used by Hoogeboom et al. (2024).

This lightweight approach enables attribute control with negligible computational overhead.

**Implementation Details.** We implement our model using GPT2-Large as the autoregressive backbone which has 770M parameters. Our unified architecture, with the two small DiT encoders, has 956M trainable parameters. During training, we use the sigmoid loss weighting and cosine noise schedule following Hoogeboom et al. (2024). For diffusion generation, we use a cosine noise schedule (Dhariwal & Nichol, 2021) with 50 diffusion steps and add small noise ($\sigma_t^2 = 0.1$) to the final generated embedding $\tilde{\mathbf{z}}_0$ to ensure robustness to minor errors introduced by diffusion sampling. We train our model on approximately 16B tokens from the FineWeb dataset (Penedo et al., 2024). We report full architecture details and hyperparameters in Appendix F.

## 4  Impact of Diffusion Process on Language Modeling

Next token prediction in traditional autoregressive models focuses primarily on local coherence, making these models inherently myopic in their generation decisions. In contrast, our diffusion-based planning approach introduces a form of non-local semantic guidance. To understand how this semantic guidance influences token prediction, we conducted a controlled experiment visualized in Figure 2. We begin with a simple prefix ("I went to the carnival and") and two distinct continuations describing different experiences. For each continuation, we compute token probabilities under two conditions: (1) when conditioned on a clean semantic embedding of the entire continuation ($z_0$) and (2) when conditioned on pure Gaussian noise ($\epsilon$). The difference in log likelihoods, $\log p(x_{cont}|x_{pref}, z_0) - \log p(x_{cont}|x_{pref}, \epsilon)$, reveals which tokens are most influenced by semantic planning with the ground-truth embedding.

As shown in Figure 2, the semantic plan primarily increases the likelihood of content-bearing words that capture the essence of each narrative. In the first continuation, words like "ring toss," "gold fish," "named," and "Flash" show the largest positive shifts. These precisely correspond to the key narrative elements describing winning a pet fish at a carnival game. Similarly, in the second continuation, words like "lost," "mirrors," "eaten," and "funnel" experience the greatest positive influence—all central to a different carnival experience involving a hall of mirrors and food. This analysis demonstrates that our model's diffusion-based planning mechanism effectively captures and emphasizes semantically meaningful content, rather than merely influencing surface-level linguistic patterns. By explicitly integrating a latent semantic plan, STAR-LDM generates coherent language that maintain global semantic consistency. We present additional experiments exploring this semantic influence across varying noise levels in Appendix A.

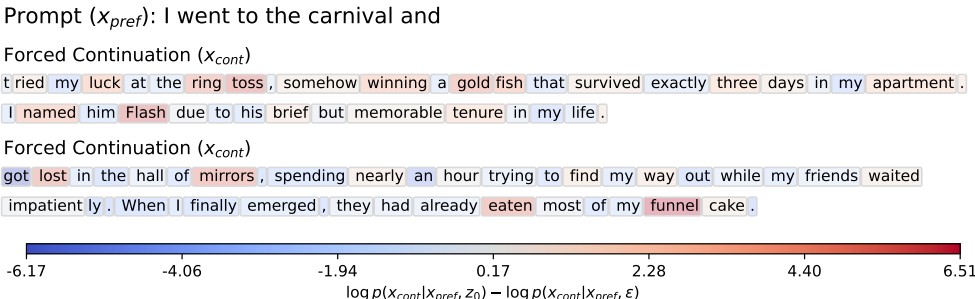

Figure 2: Visualization of the change in log likelihood due to conditioning on the clean continuation embedding versus pure noise. Tokens highlighted in red represent the information provided by the semantic plan.

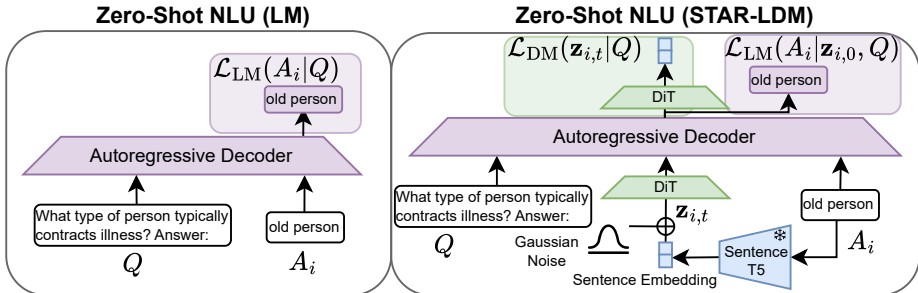

Figure 3: Diagram comparing zero-shot NLU evaluation for standard LMs (left) and STAR-LDM (right). Both models score candidate answers ($A_i$) given a question ($Q$). STAR-LDM utilizes both the answer text $A_i$ and its latent embedding $\mathbf{z}_{i,0}$ from Sentence-T5 for scoring.

## 5 Natural Language Understanding

Standard Natural Language Understanding (NLU) benchmarks, particularly multiple-choice question answering (QA) datasets (e.g., CSQA, SIQA, ARC), assess a model's ability to identify the most plausible answer or completion ($A$) from a set of candidates, given a context or question ($Q$). Conventional autoregressive models ( Figure 3, Left Panel) typically score each candidate answer $A_i$ by its negative log-likelihood: $\mathcal{L}_{\text{LM}}(A_i|Q) = -\log p(A_i|Q)$. This likelihood is calculated using teacher forcing, and the candidate with the lowest score (highest probability) is selected as the answer.

We adapt this protocol for STAR-LDM to select the most likely candidate answer given the question $Q$ (Figure 3, Right Panel). We assess candidate answers by approximating $-\log p(A_i|Q)$ using the variational lower bound on the likelihood (Kingma et al., 2021), as direct calculation is intractable. We encode each candidate $A_i$ using Sentence T5 to produce a semantic latent plan $\mathbf{z}_{i,0}$, add Gaussian noise to generate $\mathbf{z}_{i,t}$ at a specific timestep $t$, and compute our scoring function derived from the Evidence Lower Bound (ELBO):

$$-\log p(A_i|Q) \leq \mathcal{L}_{\text{LM}}(A_i|\mathbf{z}_{i,0}, Q) + \mathcal{L}_{\text{DM}}(\mathbf{z}_{i,t}|Q) \tag{10}$$

$$= -\sum_{j=1}^{|A_i|} \log p(A_i^j|A_i^{<j}, \mathbf{z}_{i,0}, Q) + \mathbb{E}_{t,\epsilon \sim \mathcal{N}(0,I)}[w_{\text{ELBO}}(\lambda_t) \cdot ||\epsilon - \epsilon_\theta(\mathbf{z}_{i,t}, t, Q)||_2^2]$$

where the weighting $w_{\text{ELBO}}(\lambda_t)$ for the regression objective follows from the derivation of the lower bound (Kingma et al., 2021). We present a detailed derivation of this scoring function in Appendix C. The LM term measures how well the model reconstructs $A_i$ given $Q$ and $\mathbf{z}_{i,0}$, while the diffusion term measures how plausible the latent plan $\mathbf{z}_{i,0}$ is for question $Q$. We select the candidate with the minimum overall score, choosing the answer that is both linguistically well-formed and has a plausible semantic meaning. We compute the expectation in Equation 10 with an unbiased Monte Carlo estimate of the weighted reconstruction error. In practice, autoregressive models often normalize the log-likelihood, $\mathcal{L}_{\text{LM}}$, by the length of the candidate $A_i$ to improve performance. We also adopt this approach as we observe it similarly improves performance.

**Results.** STAR-LDM outperforms similarly sized autoregressive models on standard zero-shot NLU benchmarks[2], as shown in Table 1. The model excels particularly on commonsense reasoning tasks (CSQA, SIQA), demonstrating the effectiveness of latent semantic planning. Notably, the diffusion component alone performs strongly, indicating that semantic plausibility contributes significantly to reasoning capabilities. While the autoregressive component underperforms independently, combining it with diffusion yields the best results, suggesting complementary strengths. This demonstrates that our diffusion planning enhances the NLU capabilities of comparable autoregressive models.

---

[2]The benchmarks include CommonsenseQA (CSQA), Social IQA (SIQA), HellaSwag (HS), Winogrande (WG), Physical IQA (PIQA), OpenBookQA (OBQA), and ARC (Easy and Challenge).

| Model | Params | Scoring | CSQA | SIQA | HS | WG | PIQA | OBQA | ARC:E | ARC:C | Avg |
|---|---|---|---|---|---|---|---|---|---|---|---|
| Random Baseline | — | | 20.0 | 33.3 | 25.0 | 50.0 | 50.0 | 25.0 | 25.0 | 25.0 | 31.7 |
| GPT2-Large | .77B | LM Loss | 36.6 | 42.1 | 42.9 | 51.9 | 69.2 | 34.2 | 46.6 | 25.1 | 43.6 |
| GPT2-XL | 1.5B | LM Loss | 36.3 | 42.3 | 47.9 | 53.1 | 70.5 | 34.4 | 51.1 | 28.5 | 45.5 |
| Pytha-1b | 1B | LM Loss | 35.3 | 42.4 | 45.6 | 52.6 | 69.2 | 32.4 | 49.1 | 27.0 | 44.2 |
| Pytha-1.4b | 1.4B | LM Loss | 35.7 | 43.2 | 50.9 | 54.2 | 71.1 | 36.0 | 54.0 | 28.3 | 46.7 |
| STAR-LDM | .96B | LM Loss | $35.2_{0.4}$ | $42.2_{0.2}$ | $44.0_{0.1}$ | $50.5_{0.1}$ | $69.3_{0.1}$ | $27.1_{0.3}$ | $37.6_{0.5}$ | $26.3_{0.3}$ | $41.5_{0.1}$ |
| | | Diffusion Loss | $48.7_{0.3}$ | $46.8_{0.5}$ | $37.9_{0.1}$ | $50.4_{0.2}$ | $65.5_{0.2}$ | $39.3_{0.1}$ | $55.0_{0.6}$ | $32.1_{0.5}$ | $47.0_{0.1}$ |
| | | LM + Diff Loss | $49.8_{0.3}$ | $46.8_{0.3}$ | $39.0_{0.1}$ | $51.4_{0.1}$ | $66.1_{0.1}$ | $41.3_{0.2}$ | $55.0_{0.3}$ | $31.4_{0.8}$ | $47.6_{0.1}$ |

Table 1: Zero Shot NLU Evaluation Results. We report mean and standard error over three monte-carlo evaluations for STAR-LDM.

## 6 StoryCloze Generation

**Experimental Setup.** To evaluate generation quality, we assess narrative continuations on the StoryCloze dataset (Mostafazadeh et al., 2016), which requires coherent fifth-sentence completions for four-sentence stories. We employ Claude 3.7 Sonnet (Anthropic, 2025) as a judge to compare STAR-LDM against baseline models (GPT-2 Large/XL and Pythia models (Biderman et al., 2023)) in blind head-to-head evaluations across 200 contexts, using four criteria: Narrative Coherence, Commonsense Reasoning, Language Quality, and Emotional/Psychological Plausibility. Full evaluation details, including the prompt, are in Appendix D.

**Results.** STAR-LDM significantly outperforms similar-sized and larger models across all criteria ( Figure 4), with win rates exceeding 70% against models like GPT-2 XL on Coherence/Reasoning ($p < 0.01$). These substantial gains suggest our latent diffusion planning effectively improves global structure and logical consistency compared to pure autoregression. While competitive against much larger models (Pythia 6.9/12B), STAR-LDM's improvements in Language Quality are less pronounced, indicating the planning mechanism primarily enhances semantic coherence rather than surface fluency.

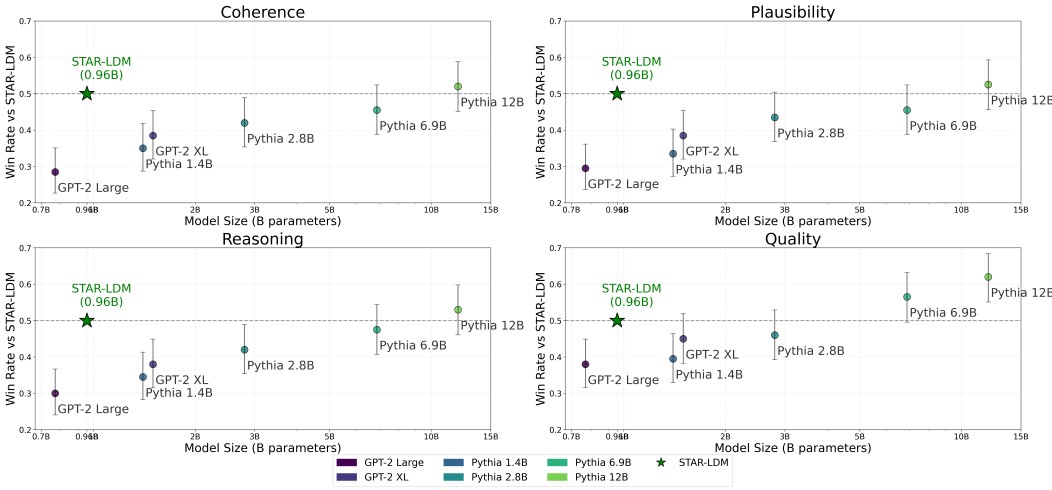

Figure 4: LLM-Judge evaluation. Results presented with 95% confidence intervals.

## 7 Language Generation Evaluation

**Experimental Setup.** We evaluate generation on 5000 C4 validation samples (Raffel et al., 2019), generating 64-token continuations from 32-token prefixes. We evaluate using three complementary metrics: (1) **Llama Ppl** measures fluency using perplexity computed by Llama-3.2-3B (Grattafiori et al., 2024); (2) **MAUVE** (Pillutla et al., 2021) assesses distributional similarity to human-written text using GPT-2 Large embeddings; and (3) **Div** quantifies

| | Params | CFG ($w$) | C4 Validation | | |
|---|---|---|---|---|---|
| | | | MAUVE ↑ | Generative Perplexity ↓ | Diversity ↑ |
| GPT-2 Large | .77B | - | $85.2_{0.6}$ | $30.5_{0.3}$ | $41.8_{0.1}$ |
| GPT-2 XL | 1.5B | - | $86.6_{0.4}$ | $27.1_{0.6}$ | $43.3_{0.1}$ |
| Pythia-1b | 1B | - | $85.5_{0.1}$ | $36.5_{0.7}$ | $45.6_{0.1}$ |
| Pythia 1.4b | 1.4B | - | $84.8_{0.5}$ | $33.1_{.1}$ | $47.1_{0.0}$ |
| STAR-LDM | .96B | 1.0 | $94.6_{0.2}$ | $36.1_{0.5}$ | $45.3_{0.0}$ |
| STAR-LDM | .96B | 1.5 | $94.1_{0.2}$ | $31.1_{0.5}$ | $45.4_{0.0}$ |
| STAR-LDM | .96B | 2.0 | $94.0_{0.2}$ | $29.8_{0.6}$ | $45.4_{0.0}$ |

Table 2: Language generation evaluation on C4 validation examples. We report mean and standard error over three sets of generations.

lexical diversity as $\prod_{n=2}^{4} \frac{|\text{unique n-grams}|}{|\text{total n-grams}|}$ (Su et al., 2022). For STAR-LDM, we report metrics across different classifier-free guidance strengths ($w$), which increase the influence of the prefix, during diffusion planning.

**Results.** As shown in Table 2, STAR-LDM consistently outperforms baselines in MAUVE score, while maintaining competitive perplexity and diversity. Increasing prefix guidance improves fluency without sacrificing diversity. We present results regarding inference latency in Appendix B.

# 8    Plug-and-Play Control

**Experimental Setup.** STAR-LDM enables attribute steering during generation without model retraining. We evaluate sentiment control and toxicity mitigation and compare against DeXperts (Liu et al., 2021a)—a strong baseline that fine-tunes specialized language models to guide generation. In contrast, our approach needs only requires training a lightweight MLP classifier on the target attribute to guide diffusion planning.

For toxicity control, we trained our MLP on the Jigsaw Unintended Bias dataset (cjadams et al., 2019). The toxicity guidance was evaluated using 1,000 randomly selected neutral prompts from RealToxicityPrompts (Gehman et al.). For sentiment control, we used both Amazon Polarity [3] and Stanford Sentiment Treebank (SST-2) (Socher et al., 2013) to train our sentiment MLP classifier. We tested sentiment steering using 1,000 randomly selected neutral prompts from OpenWebText (Gokaslan & Cohen, 2019; Liu et al., 2021a).

For evaluation in both settings, we generated 10 continuations per prompt. We measure the alignment with the control condition, language fluency, and language diversity. For toxicity, we follow prior work Deng & Raffel (2023); Liu et al. (2021a)and use the Perspective API. For sentiment we utilize RoBERTa-Large[4] (Liu et al., 2020) fine-tuned on sentiment classification across diverse domains. For language fluency, we report the generative perplexity with Llama-3.2-3B. To quantify generation diversity, we follow prior work (Lu et al., 2022b) and measure the average number of unique 3-grams in each set of continuations.

**Results.** Figure 5 shows STAR-LDM achieves better control-fluency trade-offs than DeXperts across guidance scales for both negative sentiment and toxicity reduction. Our model generates near-zero positive sentiment text at lower perplexity than GPT-2 Large, while maintaining lexical diversity even under strong guidance (additional results presented in Appendix E).

---

[3] https://huggingface.co/datasets/amazon_polarity
[4] https://huggingface.co/siebert/sentiment-roberta-large-english

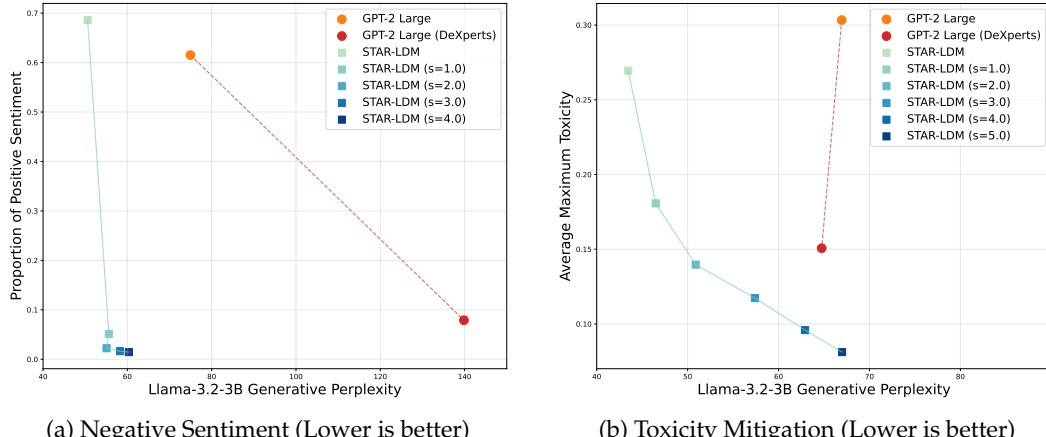

(a) Negative Sentiment (Lower is better)          (b) Toxicity Mitigation (Lower is better)

Figure 5: Relationship between perplexity and content attributes across guidance scales (*s*).

## 9   Related Work

Prior work explores both discrete diffusion (applying noise to token sequences (Hoogeboom et al., 2021; Austin et al., 2021; Hoogeboom et al., 2022; Lou et al., 2024; Sahoo et al., 2024)) and continuous approaches (in embedding space (Li et al., 2022; Gong et al., 2022; Strudel et al., 2022; Gulrajani & Hashimoto, 2023) or latent spaces (Lovelace et al., 2023; Zhang et al., 2023; Lovelace et al., 2024)). While discrete models lag behind autoregressive transformers (Zheng et al., 2024), continuous models show promise for controllability but face integration challenges that our unified architecture addresses. Controlled generation methods include model fine-tuning (domain-adaptive pretraining (Gururangan et al., 2020), control tokens (Lu et al., 2022b), RLHF (Wu et al., 2023; Ouyang et al., 2022; Jang et al., 2023)) and guided decoding with lightweight components (classifier guidance (Dathathri et al., 2019; Yang & Klein, 2021; Deng & Raffel, 2023) or specialized LMs (Krause et al., 2021; Liu et al., 2021a)). Our approach performs guidance during diffusion planning for semantic control.

Recent work enhances LM capabilities through natural language reasoning (chain-of-thought prompting (Wei et al., 2022; Zelikman et al., 2022; 2024)), specialized tokens for paused thinking (Goyal et al.), recurrent computation over hidden states (Hao et al., 2024; Geiping et al., 2025)), or by learning discrete latent actions via inverse dynamics models (Jia et al., 2025). Our work introduces "thinking" through diffusion in semantic space, complementing discrete token generation. Hybrid generative architectures have been explored in prior work primarily for multimodal settings rather than enhancing language generation alone (Zhou et al., 2024; Wang et al., 2024; Team, 2024; Liang et al., 2024).

## 10   Conclusion

We presented STAR-LDM, a unified architecture that integrates autoregressive language modeling with latent diffusion planning. Our experiments demonstrate three key advantages: (1) improved natural language understanding capabilities, (2) higher quality narrative generation with better coherence and reasoning, and (3) effective plug-and-play control without model fine-tuning. By enabling models to "stop and think" in a continuous semantic space before generating text, STAR-LDM combines the global planning capabilities of diffusion models with the fluency of autoregressive generation. This approach not only improves performance on standard tasks but also enables precise control of generated text attributes through lightweight guidance mechanisms. Our work represents a step toward language models that more closely mirror human writing processes, pausing to plan before committing to specific wording.

## Acknowledgments

CB is supported by the National Science Foundation (NSF) through the NSF Research Traineeship (NRT) program under Grant No. 2345579. SZ is supported by the Defense Advanced Research Projects Agency (DARPA) under Grant No. D24AP00259-00. This research is supported by grants from the National Science Foundation NSF (OAC- 2118310, OAC-1934714, IIS-2107161, and IIS-1724282, HDR-2118310), the Cornell Center for Materials Research with funding from the NSF MRSEC program (DMR-1719875), DARPA, arXiv, and the New York-Presbyterian for the NYP-Cornell Cardiovascular AI Collaboration. The authors acknowledge the National Artificial Intelligence Research Resource Pilot (NAIRR-240157) and TACC Vista for contributing to this research result.

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

## A Additional Visualizations of the Impact of Diffusion Process on Language Modeling

We provide a number of additional results in this section to help build intuition for the contribution of the diffusion model to language generation. Both Figure 2 and Figure 6, show the change in log likelihood at a token level when conditioned on a semantic plan. Figure 2 shows this for arbitrary text continuations which we embed as semantic plans with Sentence T5, while Figure 6 shows this for latent plans sampled from STAR-LDM.

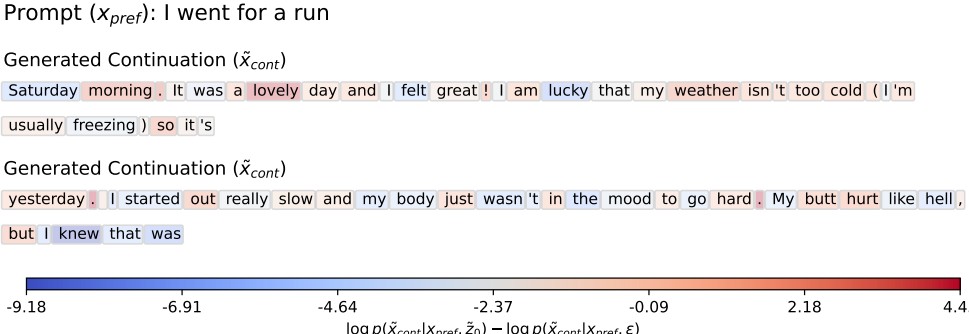

Figure 6: Visualization of the change in log likelihood due to conditioning on a semantic plan. Shown for two different generations. Positive values (red) indicate tokens that become more likely when conditioned on the plan, while negative values (blue) indicate tokens that become less likely.

In order to quantitatively assess the impact of the semantic plans, we measure the perplexity of STAR-LDM as a function of the log(SNR) in Figure 7. We do this on a subset of the C4 validation set, as well as on a set of generations nucleus sampled from STAR-LDM. We see that at high log(SNR), where the semantic plan is very clean, the perplexity is much lower. However at low log(SNR), where the semantic plan is very noisy, the perplexity is much higher. This demonstrates that the semantic plan reduces the uncertainty in the language generation; helping the language model focus on a specific semantic direction.

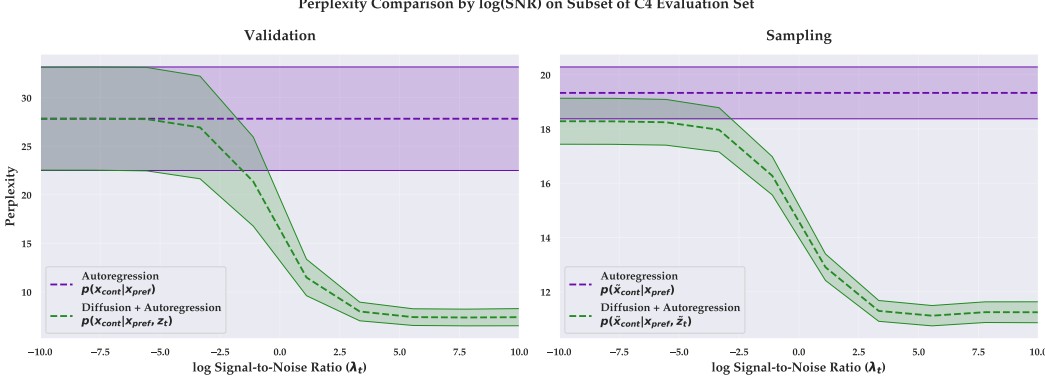

Figure 7: Perplexity at various noise levels on a subset of 100 examples from the C4 evaluation set (left), or 100 sampled generations (right).

Additionally we show the same perplexity comparison but at the maximum log(SNR), corresponding to $t = 0$, in Figure 8.

Finally to complement our quantitative evaluation of STAR-LDM across noise levels, we visualize how the token wise log likelihoods change with the log(SNR) in Figure 9.

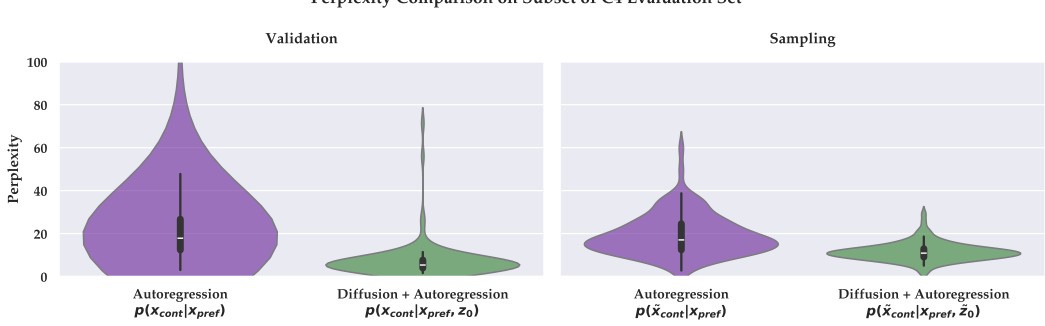

Figure 8: Perplexity on a subset of 100 examples from the C4 evaluation set (left), or 100 sampled generations (right).

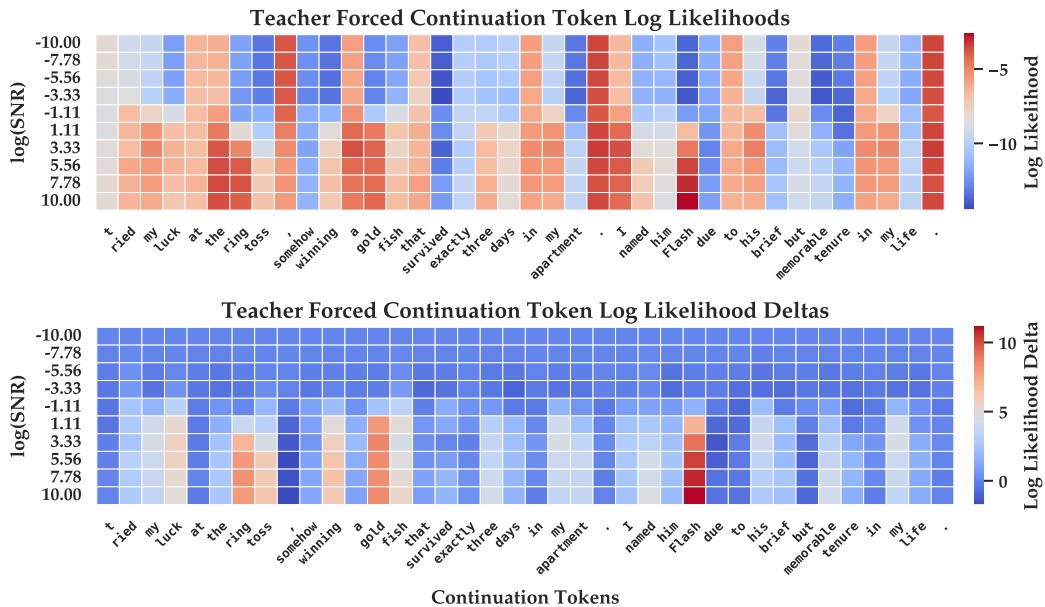

Figure 9: Visualization of the token wise log likelihood by log(SNR) (top). Visualization of the token wise change in log likelihood by log(SNR) (bottom).

## B   Inference Latency

To provide a comprehensive view of the practical trade-offs of our method, we analyze the inference latency of STAR-LDM. As the primary source of computational overhead is the iterative "thinking" phase, we evaluate how generation quality varies with the number of diffusion sampling steps. For this analysis, we adopt the DPM-Solver++(2M) SDE sampler (Lu et al., 2022a), which is highly effective for few-step sampling, and we sweep the number of steps from 5 to 50. We set the prefix guidance to $w = 2.0$ for this study.

We generate 96-token continuations for 32-token prompts, using a set of 250 examples randomly drawn from the C4 validation set. We report generative perplexity and diversity as a function of the wall-clock time per generation in Figure 10. For reference, we also benchmark the inference speed and quality of the standard GPT-2 Large and GPT-2 XL autoregressive models.

It is important to note that our current STAR-LDM implementation is not optimized for inference speed. Specifically, key-value (KV) caching, a standard technique for accelerating autoregressive generation, is not yet implemented for the diffusion planning phase, during

which the prefix is re-processed at each denoising step. Our reported wall-clock times for STAR-LDM are therefore a loose upper bound on its inference speed.

As shown in Figure 10, STAR-LDM introduces a tunable trade-off between computation time and generative perplexity. We observe diminishing returns as the number of sampling steps increases; perplexity improves up to around 15-20 steps and then largely plateaus, while latency continues to increase. In that sampling step range, the latency is similar to GPT-2 XL, which STAR-LDM consistently outperforms in NLU evaluations and generative metrics such as MAUVE score.

Figure 10: **Quality-Speed Tradeoff in Language Generation.** We plot generative perplexity (lower is better) against the wall-clock time to generate a 96-token continuation. The STAR-LDM curve shows performance as the number of diffusion sampling steps is varied. Baselines are standard autoregressive models.

# C  NLU Derivation

In this appendix, we derive the Evidence Lower Bound (ELBO) for our STAR-LDM model, which serves as the foundation for both training and evaluation. This derivation illustrates how standard diffusion ELBO principles are adapted when the reconstruction term is parameterized with an autoregressive decoder.

## C.1  Standard Diffusion ELBO

We first recall the standard diffusion ELBO for continuous diffusion models (Kingma et al., 2021). For a diffusion model that generates data starting from pure noise, the log-likelihood of data point $x$ can be bounded from below as:

$$\log p(x) \geq \mathcal{L}_{\text{ELBO}}(x) = \underbrace{\mathbb{E}_{q(z_{\lambda_{\max}}|x)}[\log p(x|z_{\lambda_{\max}})]}_{\text{Reconstruction term}} \tag{11}$$

$$- \underbrace{D_{\text{KL}}(q(z_{\lambda_{\min}}|x) \parallel p(z_{\lambda_{\min}}))}_{\text{Prior matching term}} \tag{12}$$

$$- \underbrace{\int_{\lambda_{\min}}^{\lambda_{\max}} \frac{1}{2}\|\epsilon - \epsilon_\theta(z_\lambda, \lambda)\|^2 d\lambda}_{\text{Diffusion loss term}} \tag{13}$$

Where: $z_\lambda$ represents the noisy latent variable at noise level parameterized by $\lambda$ (log-SNR); $q(z_\lambda|x)$ is the forward process that adds noise to the data; $p(x|z_{\lambda_{\max}})$ is the reconstruction term (generating data from noisy latent); $D_{\text{KL}}(q(z_{\lambda_{\min}}|x) \parallel p(z_{\lambda_{\min}}))$ is the prior matching term; and the integral term represents the diffusion denoising objective.

## C.2 ELBO for STAR-LDM

For STAR-LDM, we adapt this framework to the language modeling context where we have a prefix $x_{\text{pref}}$ and continuation $x_{\text{cont}}$. The key innovation is that the reconstruction term is parameterized using an autoregressive decoder. We derive the ELBO for $p(x_{\text{cont}}|x_{\text{pref}})$ as follows:

First, we introduce the latent sentence embedding $z$ that represents the semantic plan for the continuation. The joint probability can be factorized as:

$$p(x_{\text{cont}}, z|x_{\text{pref}}) = p(x_{\text{cont}}|x_{\text{pref}}, z)p(z|x_{\text{pref}}) \tag{14}$$

The marginal probability of the continuation is:

$$p(x_{\text{cont}}|x_{\text{pref}}) = \int p(x_{\text{cont}}|x_{\text{pref}}, z)p(z|x_{\text{pref}})dz \tag{15}$$

Since this integral is intractable, we derive a lower bound using variational inference. We introduce a variational posterior $q(z|x_{\text{cont}})$ which in our case is implemented by the Sentence-T5 encoder which produces a clean embedding, which then serves as the basis for the forward diffusion process.

Applying the standard variational inference procedure and Jensen's inequality, we arrive at the Evidence Lower Bound. Here, $z$ represents the full hierarchy of latent variables defined by the diffusion process::

$$\log p(x_{\text{cont}}|x_{\text{pref}}) \geq \mathbb{E}_{q(z|x_{\text{cont}})}\left[\log p(x_{\text{cont}}|x_{\text{pref}}, z)\right] - D_{\text{KL}}(q(z|x_{\text{cont}}) \parallel p(z|x_{\text{pref}})) \tag{16}$$

For the diffusion model part, we expand the KL term using its standard parameterization. The reconstruction term $p(x_{\text{cont}}|x_{\text{pref}}, z)$ is evaluated at a specific low-noise level $\lambda_{\max}$, which corresponds to the latent variable $z_{\lambda_{\max}}$. The KL divergence over the full diffusion path from the variational posterior $q(z|x_{\text{cont}})$ to the prefix-conditioned prior $p(z|x_{\text{pref}})$ can then be expressed as:

$$D_{\text{KL}}(q(z|x_{\text{cont}}) \parallel p(z|x_{\text{pref}})) = D_{\text{KL}}(q(z_{\lambda_{\min}}|x_{\text{cont}}) \parallel p(z_{\lambda_{\min}})) \tag{17}$$

$$+ \int_{\lambda_{\min}}^{\lambda_{\max}} \frac{1}{2}\mathbb{E}_{q(z_\lambda|x_{\text{cont}})}[\|\epsilon - \epsilon_\theta(z_\lambda, \lambda, x_{\text{pref}})\|^2]d\lambda \tag{18}$$

The first term is the KL divergence between the prior and the most noisy distribution. Substituting this back into (16), our final, complete ELBO is:

$$\log p(x_{\text{cont}}|x_{\text{pref}}) \geq \mathbb{E}_{q(z_{\lambda_{\max}}|x_{\text{cont}})} \left[ \log p_\theta(x_{\text{cont}}|x_{\text{pref}}, z_{\lambda_{\max}}) \right]$$

$$- D_{\text{KL}}(q(z_{\lambda_{\min}}|x_{\text{cont}}) \parallel p(z_{\lambda_{\min}})) \tag{19}$$

$$- \int_{\lambda_{\min}}^{\lambda_{\max}} \frac{1}{2} \mathbb{E}_{q(z_\lambda|x_{\text{cont}})}[\|\epsilon - \epsilon_\theta(z_\lambda, \lambda, x_{\text{pref}})\|^2]d\lambda \tag{20}$$

The prior matching term (19) ensures that the distribution of the most-noised latents matches a standard Gaussian prior. In many practical applications, this term is either assumed to be approximately zero (if the noise schedule is such that $q(z_{\lambda_{\min}}|x_{\text{cont}})$ collapses to a standard normal) or is omitted from the training objective as it is often small and constant with respect to the model parameters $\theta$.

### C.3 Parameterization with Autoregressive Decoder

For STAR-LDM, we parameterize the reconstruction term with an autoregressive decoder. Specifically:

$$\log p_\theta(x_{\text{cont}}|x_{\text{pref}}, z) = \sum_{i=1}^{|x_{\text{cont}}|} \log p_\theta(x_{\text{cont},i}|x_{\text{pref}}, x_{\text{cont},<i}, z) \tag{21}$$

This allows the model to generate text token-by-token while being guided by the semantic plan represented by the latent variable $z$.

### C.4 ELBO for Zero-Shot Evaluation

For zero-shot NLU evaluation, we use the negative of the ELBO derived above to score candidate answers $A_i$ given a question $Q$. For the purpose of creating a practical scoring function, we drop the prior matching term ($D_{\text{KL}}(...)$ in (19)). This is a common practice as the term is constant with respect to the model parameters $\theta$ being evaluated and does not affect the relative ranking of candidate answers. Its non-negativity ensures that we are still optimizing a valid (though slightly looser) bound.

A key aspect of our evaluation protocol is that the reconstruction noise level, $\lambda_{\max}$, is a tunable hyperparameter. This allows us to precisely control the trade-off between the influence of the semantic plan (the diffusion component) and the fluency of the autoregressive decoder.

The resulting scoring function is based on the two main components of the ELBO bound: the reconstruction loss and the diffusion loss.

$$\text{Score}(A_i|Q; \lambda_{\max}) = -\mathbb{E}_{q(z_{i,\lambda_{\max}}|A_i)} \left[ \sum_{j=1}^{|A_i|} \log p_\theta(A_i^j|A_i^{<j}, z_{i,\lambda_{\max}}, Q) \right] \tag{22}$$

$$+ \mathbb{E}_{t,\epsilon \sim \mathcal{N}(0,I)} \left[ w_{\text{ELBO}}(\lambda_t) \cdot \|\epsilon - \epsilon_\theta(z_{i,t}, t, Q)\|^2 \right] \tag{23}$$

In practice, we compute an unbiased estimate of this score by approximating both expectations with Monte Carlo sampling[5].

In this formulation, $z_{i,\lambda_{\max}}$ represents the candidate answer's clean embedding noised to the level $\lambda_{\max}$, while $z_{i,t}$ is the same embedding noised to a random level $t$. By varying $\lambda_{\max}$, we control how much the autoregressive decoder relies on the semantic plan versus its own

---

[5]We use 16 Monte Carlo samples for the reconstruction term and 128 for the diffusion term, which we found sufficient for stable estimates.

internal knowledge. We fix $\lambda_{\max} = -2$ across all datasets, though we observed that tuning this on a per-dataset basis would lead to further improvements. The candidate answer with the lowest score is selected as the most probable answer.

This derivation provides the theoretical foundation for the zero-shot evaluation protocol for STAR-LDM.

## D   StoryCloze Evaluation Details

For our Claude judged StoryCloze evaluation we randomly sample a subset of 200 stories from the StoryCloze validation split. For each story in our sample, we generate a 32-token continuation from each baseline as well as STAR-LDM. We then compare for each story each baseline generation against that of STAR-LDM by inserting the two continuations into the prompt show in Figure 11. The ordering of the continuations (i.e. which is given as option A/B) is randomized for every comparison. The prompt gives explicit instruction on the criteria to use to evaluate the continuations as well as how to denote the winner. The winner is indicated with specific formatting that is easily parsed with a regular expression. We present a qualitative example in Figure 12.

## E   Additional Control Experiments

In this appendix, we provide additional results for our controlled text generation experiments that complement the findings presented in Section 7.

### E.1   Control-Perplexity Trade-offs

While the main paper shows negative sentiment control and the average maximum toxicity metric, here we present additional control-perplexity trade-offs for positive sentiment and the average toxicity metric.

Figure 13 demonstrates that STAR-LDM effectively controls positive sentiment and reduces average toxicity scores while maintaining competitive perplexity. For positive sentiment control, our model achieves high scores ($> 0.95$) with guidance scale $s = 2.0$, showing minimal fluency degradation compared to the base model. Similarly, for toxicity reduction, STAR-LDM achieves substantially lower average toxicity scores than both GPT-2 Large and DeXperts at comparable perplexity levels.

### E.2   Control-Diversity Trade-offs

We also evaluate the relationship between attribute control and lexical diversity, measured by the ratio of unique trigrams in the generated output. These metrics complement the perplexity evaluations by assessing whether controlled generation maintains vocabulary richness and avoids repetitive patterns.

Figures 14 and 15 show that STAR-LDM consistently outperforms DeXperts by achieving better attribute control while maintaining higher lexical diversity. For both positive and negative sentiment control, our approach preserves trigram diversity even at strong guidance scales. Similarly, for toxicity mitigation, STAR-LDM with guidance scale $s = 5.0$ dramatically reduces both average and maximum toxicity while maintaining diversity comparable to the base model.

These results highlight a key advantage of our diffusion planning approach: by guiding generation at the semantic level before committing to specific tokens, STAR-LDM achieves natural attribute control that preserves linguistic richness without resorting to repetitive or restricted vocabulary choices.

---

### LLM-as-a-Judge Prompt

**Task Overview**
You are evaluating two competing language models that generate story continuations. Given a four-sentence story context, each model will generate a continuation. Your task is to judge which continuation is better based on a set of defined criteria.

**Evaluation Criteria**

**Narrative Coherence** How well does the continuation logically follow from the context? Does it maintain consistency with characters, events, and situations established in the first four sentences? Does it advance the narrative in a natural and satisfying way?

**Commonsense Reasoning** Does the continuation demonstrate understanding of causal and temporal relationships? Is it consistent with real-world knowledge about how events typically unfold? Does it avoid contradicting established facts about the world?

**Language Quality** Is the continuation grammatically correct and well-formed? Does it match the style, tone, and vocabulary level of the context? Does it flow naturally from the preceding text?

**Emotional/Psychological Plausibility** Does the continuation reflect plausible human reactions, motivations, or emotions? Does it appropriately address characters' goals, feelings, or intentions set up in the context?

**Scoring Process** Read the four-sentence context carefully. Read both generated continuations (labeled A and B) without knowing which system produced which text. Rate each continuation on all four criteria using a 1-5 scale. For each criteria select the continuation with the higher score as the winner. Provide a brief justification (2-3 sentences) explaining your decision.

Clearly indicate your final judgment for each criteria by placing the criteria (either "Coherence" or "Reasoning" or "Quality" or "Plausibility") and winning continuation (either "A" or "B") within double brackets, like: [[Coherence A]]. Make sure to provide a winner for each criteria. Missing Judgements for Criteria will be considered a tie.

**Example Evaluations**

**Example 1:**
**Context:** Bob was bored at his job as a school teacher. He had been working so hard this past month. He decided to treat himself with something special. He ordered tickets for a weekend snowboarding trip.

**Continuation A:** He was looking forward to getting away.
**Continuation B:** His boss told him he had to work this weekend.

**Judgement: Narrative Coherence:** Continuation A (4/5) follows the established narrative arc of Bob's self-reward better than Continuation B (3/5). A naturally builds on his anticipation while B introduces a conflicting element that shifts away from the focus on Bob's well-deserved break. [[Coherence A]] **Commonsense Reasoning:** Continuation A (5/5) demonstrates better understanding of the natural progression from planning a trip to looking forward to it, while B (4/5) introduces a plausible but less aligned obstacle that breaks the causal flow of treating oneself. [[Reasoning A]] **Language Quality:** Both continuations are grammatically sound and match the context's tone, with A (4/5) flowing slightly more naturally than B (4/5) as it maintains the focus on Bob's perspective rather than shifting to his boss's actions. [[Quality A]] **Emotional/Psychological Plausibility:** Continuation A (4/5) better addresses Bob's established intention to reward himself with positive anticipation, while B (4/5), though realistic, introduces disappointment that contradicts the emotional trajectory set up in the context. [[Plausibility A]]

**Example 2:**
**Context:** Olivia went out with Harry on a date. Harry thought the date was going well. Olivia thinks he is a complete jerk and never wants to see him again. Harry keeps raving about their chemistry.

**Continuation A:** Olivia is about to leave.
**Continuation B:** Olivia had her friend call her, to tell her that her mother was ill so as not to offend him.

**Judgement: Narrative Coherence:** Continuation B (5/5) provides a more specific action that builds upon Olivia's established discomfort and desire to end the date without confrontation, while A (4/5) is logical but less developed. [[Coherence B]] **Commonsense Reasoning:** Continuation B (5/5) demonstrates better understanding of complex social dynamics and face-saving strategies commonly used in uncomfortable dating situations, compared to A (4/5) which is direct but lacks social nuance. [[Reasoning B]] **Language Quality:** Both continuations are grammatically correct, but B (4/5) provides more detail matching the context's complexity, while A (3/5) is much simpler than the established narrative style. [[Quality B]] **Emotional/Psychological Plausibility:** Continuation B (5/5) shows deeper understanding of Olivia's conflicting desires to escape while avoiding direct confrontation, consistent with her awareness of Harry's obliviousness, while A (4/5) captures her desire to leave but not her likely approach. [[Plausibility B]]

**Example 3:**
**Context:** Jack and Ferris always fought for headphones. One day Jack broke Ferris' headphones while jogging. Ferris was furious at Jack. Their parents yelled at them.

**Continuation A:** Jack promised Ferris to buy him new headphones.
**Continuation B:** Jack promised to take Ferris jogging.

**Judgement: Narrative Coherence:** Continuation A (5/5) directly addresses the central conflict (broken headphones) with an appropriate resolution, while B (2/5) ignores the primary issue established in the context. [[Coherence A]] **Commonsense Reasoning:** Continuation A (5/5) offers a logical resolution consistent with real-world problem-solving, while B (2/5) fails by suggesting the very activity that caused the problem, which would likely worsen the situation. [[Reasoning A]] **Language Quality:** Both continuations are grammatically correct, but A (4/5) flows more naturally from the context's focus on the headphone conflict, while B (4/5) creates a jarring shift in focus. [[Quality A]] **Emotional/Psychological Plausibility:** Continuation A (5/5) acknowledges responsibility and offers appropriate compensation addressing Ferris's established anger, while B (3/5) shows poor understanding of the emotional situation and would likely increase Ferris's frustration. [[Plausibility A]]

**Implementation Notes** Judge continuations primarily on how well they follow from the established context. Prioritize narrative coherence and commonsense reasoning over creativity or surprise. Be consistent in applying the evaluation criteria across different story types. Consider the natural progression of events and character motivations when evaluating plausibility. Avoid bias toward specific narrative styles or genre preferences.

Figure 11: Prompt template used for LLM-as-Judge evaluations with Claude 3.7 Sonnet on StoryCloze generations.

---

**Context:** Rick grew up in a troubled household. He never found good support in family, and turned to gangs. It wasn't long before Rick got shot in a robbery. The incident caused him to turn a new leaf.

---

**STAR-LDM (A):** Ricks attempt to change his life was long and hard. However, the process has been worth it. The way he lived changed when he started a new chapter...

---

**GPT2-XL (B):** Rick now looks after his younger brother, who is disabled.

The two are the heart of what is perhaps the most important project this summer: a massive...

---

**Evaluation of Story Continuations**
After analyzing the four-sentence context about Rick and the two competing continuations, here is my assessment based on the specified criteria:
**Narrative Coherence**
Continuation A follows the context's redemption arc naturally, describing Rick's difficult but rewarding journey to change his life after the shooting incident. Continuation B introduces Rick caring for a disabled younger brother (new information) and then abruptly shifts to discussing a "project" without clear connection to the established narrative. A maintains consistent focus on Rick's transformation while B introduces disconnected elements.
**[[Coherence A]]**
**Commonsense Reasoning**
Continuation A demonstrates understanding of how personal transformation is typically difficult but rewarding, showing appropriate causal relationships. Continuation B presents a plausible caregiving scenario but fails to connect this to the "important project" mentioned afterward, breaking logical flow. The second sentence in B feels disconnected from both the context and the first sentence of the continuation.
**[[Reasoning A]]**
**Language Quality**
Continuation A has minor issues (e.g., "Ricks" missing apostrophe) but maintains the narrative's tone and flows naturally from the context. Continuation B starts cohesively but abruptly transitions to an incomplete thought about a "project," creating a jarring stylistic break that doesn't match the established narrative voice. A feels more like a complete thought while B seems fragmented.
**[[Quality A]]**
**Emotional/Psychological Plausibility**
Continuation A acknowledges the psychological reality that changing one's life after trauma is challenging but worthwhile, reflecting plausible emotional growth. Continuation B presents a plausible caring relationship but fails to connect this to Rick's emotional journey established in the context, and the sudden mention of a "project" lacks psychological grounding in the character's established motivations.
**[[Plausibility A]]**

---

Figure 12: Qualitative example of story completions from the StoryCloze dataset. The LLM judge evaluated both continuations, where Continuation A is STAR-LDM and Continuation B is GPT2-XL. The judge preferred STAR-LDM's continuation across all evaluation criteria.

# F Implementation Details

## F.1 STAR-LDM

We report additional architecture details and hyperparemeters for STAR-LDM in Table 3. Our architecture processes the noisy latent embedding through a sequence of two DiT blocks, with the main autoregressive decoder situated in between (see Figure 1). To enable classifier-free guidance for the prefix, we must simultaneously train our model to perform unconditional prediction, i.e. without information from the prefix. We achieve this with two modifications. First, to remove the prefix conditioning, we replace the output representations

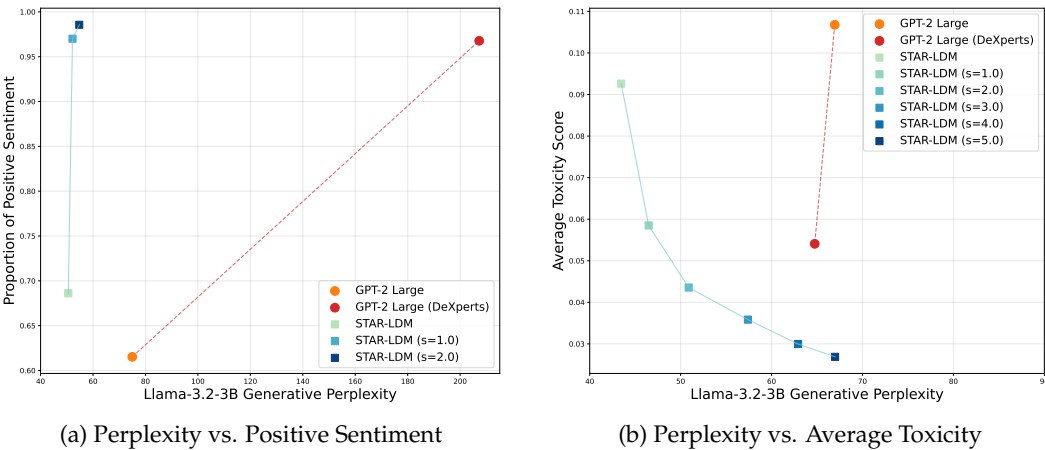

(a) Perplexity vs. Positive Sentiment          (b) Perplexity vs. Average Toxicity

Figure 13: Additional control-perplexity trade-offs beyond those shown in the main paper.

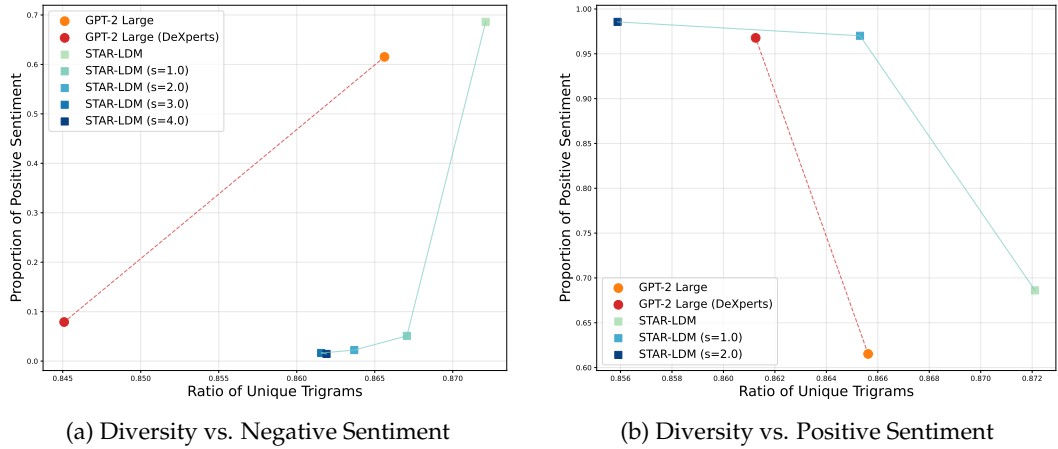

(a) Diversity vs. Negative Sentiment          (b) Diversity vs. Positive Sentiment

Figure 14: Relationship between n-gram diversity and sentiment control.

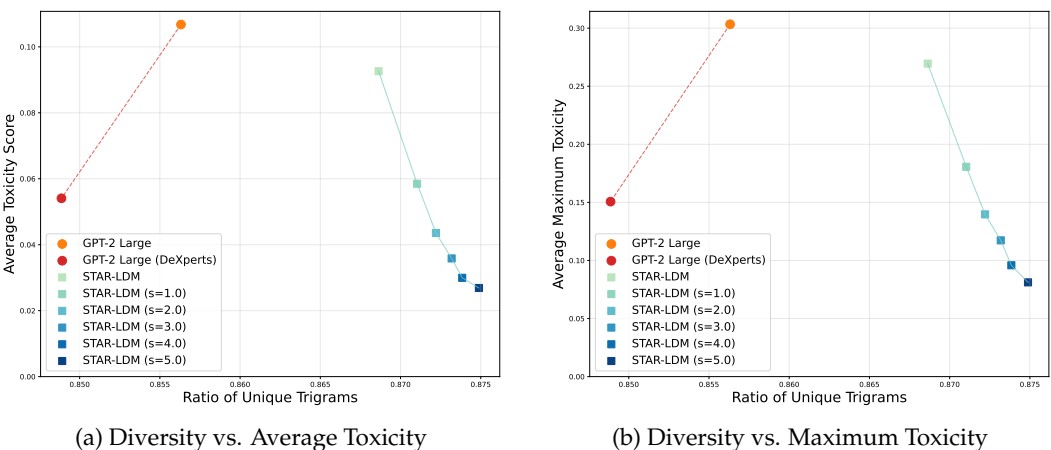

(a) Diversity vs. Average Toxicity          (b) Diversity vs. Maximum Toxicity

Figure 15: Relationship between n-gram diversity and toxicity control.

Table 3: Implementation details for STAR-LDM training.

| | |
|---|---|
| **Architecture** | |
| Base Language Model | GPT-2 Large (774M parameters) |
| Sentence Embedding Model | Sentence-T5 XL (768 dimensions) |
| Total Trainable Parameters | 956M |
| **DiT Architecture — Prompt Encoder and Diffusion Prediction** | |
| DiT Layers per Module | 6 |
| DiT Hidden Dimension | 1024 |
| DiT Attention Heads | 16 |
| DiT Head Dimension | 64 |
| Activation Function | SwiGLU (Shazeer, 2020) |
| Normalization Layer | Adaptive RMSNorm (Zhang & Sennrich, 2019; Peebles & Xie, 2022) |
| Soft Prompt Sequence Length | 8 |
| **Diffusion Configuration** | |
| Output Parameterization | v-prediction (Salimans & Ho, 2022) |
| Diffusion Steps (Inference) | 50 |
| Diffusion Sampler (Inference) | DDPM (Ho et al., 2020) |
| Loss Weighting | Sigmoid (Hoogeboom et al., 2024) |
| Noise Schedule | Cosine (Dhariwal & Nichol, 2021) |
| **Training Details** | |
| Dataset | FineWeb |
| Training Tokens | 16B |
| Max Sequence Length | 128 |
| Optimizer | AdamW (Loshchilov & Hutter, 2019) |
| Learning Rate | 5e-4 |
| Training Steps | 250000 |
| Batch Size | 512 |
| Weight Decay | .01 |
| Gradient Clipping | 1.0 |
| Diffusion Loss Weight ($\beta$) | 5.0 |

from the autoregressive decoder with a single, learnable null-context embedding. However, this embedding contains no information about the noisy sentence embedding processed by the first DiT block. To preserve this initial processing, we therefore introduce a UNet-style skip connection (Ronneberger et al., 2015) that concatenates the output of the first DiT directly to the null-context embedding before it is passed to the second DiT. This design ensures the unconditional path still benefits from the initial latent processing while remaining independent of the prefix. For the conditional path, the output of the first DiT is concatenated with the output of the autoregressive backbone, integrating information from the prefix.

## F.2 Decoding Parameters

For all of our generation experiments, we utilize nucleus sampling (p=0.95) (Holtzman et al.) with a repetition penalty of 1.2 for the STAR-LDM decoder and the autoregressive baselines.

## F.3 Baseline NLU Results

All baseline results are computed with the lighteval evaluation harness[6] using the FineWeb (Penedo et al., 2024) evaluation prompts[7].

---

[6]https://github.com/huggingface/lighteval
[7]https://huggingface.co/datasets/HuggingFaceFW/fineweb/blob/main/lighteval_tasks.py

Table 4: Implementation details for the Noise-Conditioned MLP classifier.

| MLP Architecture | |
|---|---|
| Input Dimension | 768 |
| Hidden Dimension | 1536 |
| Number of MLP Blocks | 4 |
| Number of Hidden Layers | 1 hidden layer |
| Normalization | Pre-norm with adaptive RMSNorm |
| Residual Connections | Applied after each MLP |
| Activation Function | SwiGLU |
| Time Embedding | Sinusoidal positional encoding |
| **Diffusion Configuration** | |
| Noise Schedule | Cosine (Dhariwal & Nichol, 2021) |
| Loss Weighting | Sigmoid (Hoogeboom et al., 2024) |
| **Training Details** | |
| Optimizer | AdamW |
| Learning Rate | 1e-4 |
| Batch Size | 256 |
| Weight Decay | .01 |
| Gradient Clipping | 1.0 |

## F.4 Noise-Conditioned MLP

We present results of the noise conditioned classifier for sentiment and toxicity detection on held-out sets for our dataset in Figure 16. We observe that they smoothly improve in performance as noise levels decrease, outperforming a logistic regression baseline for even moderately noisy data.

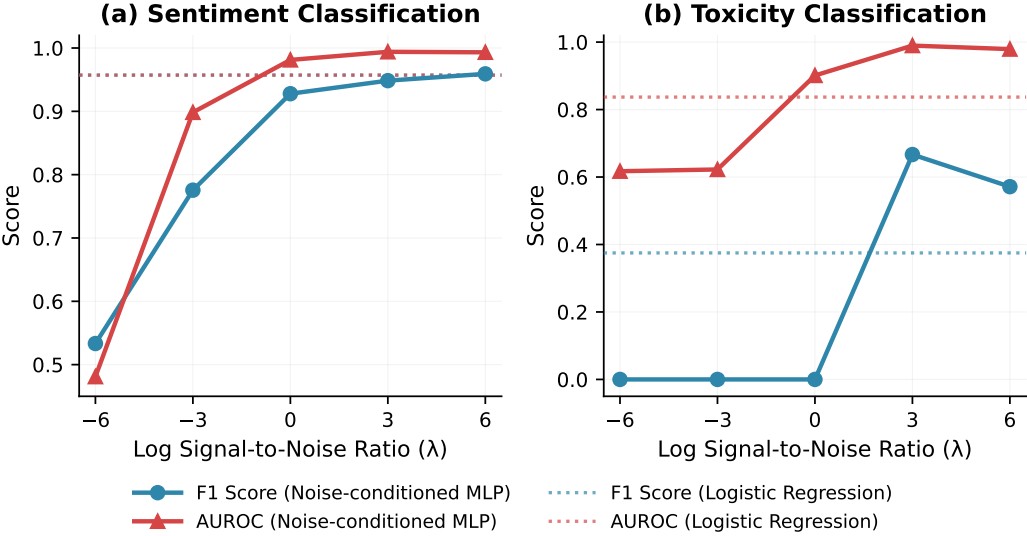

Figure 16: Classifier performance across noise levels for sentiment (left), and toxicity (right) classification.

