# OpenReview forum: "Stop-Think-AutoRegress: Language Modeling with Latent Diffusion Planning"
_colmweb.org/COLM/2025/Conference — COLM 2025_

### Official Review · Reviewer_fPq9 · 2025-05-11

**Rating:** 6
**Confidence:** 4
**Ethics Flag:** 1

**Summary:**

This paper proposes Stop-Think-AutoRegress Language Diffusion Model (STAR-LDM), which integrates latent diffusion planning with autoregressive generation.

**Questions To Authors:**

1. The biggest model used in experiments is 1.5B. How about scaling model size to 7B?

2. The compared model does not inherently have thinking ability. You can consider comparing with thinking Models like Qwen3.

3. A related work, "Controlling Large Language Model with Latent Actions", can be discussed.

4. The incorporated diffusion module learns to model the embedding distribution from Sentence-T5. However, the autoregressive model backbone is GPT2-Large. Why does the embedding of Sentence-T5 can effectively manipulate the generation process of GPT2-Large? Moreover, Sentence-T5 may not be strong enough if we use a larger autoregressive backbone.

**Reasons To Accept:**

The writing is clear and easy to follow.

**Reasons To Reject:**

1. To train the diffusion model, we need embeddings from another model (e.g., the used Sentence-T5). But authors do not discuss the impact of this embedding model.

2. Lacking experiments on larger models.

---

> ### Author Response · Authors · 2025-06-03
>
> We thank Reviewer fPq9 for noting our paper's clarity and for their insightful questions regarding model components and comparisons.
>
> **Re: Impact & Choice of Sentence-T5 Embeddings**
>
> During training (Sec. 3.1, Fig. 1), our GPT2-Large autoregressive decoder is conditioned on a noisy Sentence-T5 XL embedding (z_t) of the ground truth continuation, processed by DiT modules into a soft prompt. The entire STAR-LDM model, including DiTs and decoder, learns via the language modeling (and diffusion) loss to use this latent semantic information to guide generation.
> Sentence embeddings are inherently lossy, capturing broad semantics but not all fine-grained detail. Information not in this plan is naturally handled by the AR decoder, which remains responsible for fluency and local dependencies. DiTs (lines 115-118) are crucial for translating between the continuous embedding space (plan) and discrete token representations.
> We chose Sentence-T5 XL as a strong, widely-used sentence encoder. While larger AR backbones might benefit from more powerful or co-adapted encoders, we selected a popular, effective model to demonstrate our architecture. Further investigation is left future work.
>
> **Re: Lack of experiments on larger models (e.g., 7B)**
>
> Our primary contribution is the methodological innovation of STAR-LDM, integrating latent diffusion planning with autoregressive generation. We demonstrated its efficacy at a scale of approximately 1B parameters, comparing against models like GPT-2 XL (1.5B) and Pythia-1.4B. Training generative models in the 1B+ parameter regime from scratch or fine-tuning them extensively requires computational resources beyond our current capacity.
>
> However, to explore scalability, we conducted preliminary experiments adapting STAR-LDM to a Llama 3.1 8B backbone. To make this feasible, we froze the Llama parameters and fine-tuned only our added DiT modules on ~16B tokens—orders of magnitude less data than Llama's 15T+ pre-training.
>
> Even under these significant constraints, this preliminary STAR-LDM (8B) showed promise, outperforming the frozen Llama 3.1 8B base on several NLU benchmarks that benefit from global semantic planning (e.g., CSQA: 59.2% vs. 47.9%; OBQA: 48.8% vs. 43.4%; SIQA: 51.6% vs. 47.7%). It underperformed on tasks potentially requiring more nuanced local reasoning from a fully trained AR component (e.g., ARC-E 73.3% vs. 82.6%; HellaSwag 46.0% vs. 76.3%). This aligns with our discussion on the complementary strengths of diffusion planning and autoregressive modeling (see response to Reviewer UtyY).
>
> The frozen LLM backbone and the vast difference in relevant training data (16B vs. 15T+ tokens) undoubtedly limited the full potential of STAR-LDM at this scale. Nevertheless, these initial findings suggest that our architecture can enhance certain capabilities even in larger models and provide valuable insights for future work on more comprehensively scaling and co-training such hybrid systems. We will briefly discuss these such directions in the final version.
>
> **Re: Comparison to "thinking" models like Qwen3**
>
> We appreciate the suggestion to compare with models like Qwen3 that exhibit "thinking" abilities. STAR-LDM's "thinking" mechanism is distinct: it involves latent diffusion planning in a continuous semantic space to generate a global plan before token-level generation. This differs from, and is somewhat orthogonal to, methods that elicit thinking through explicit token-space chain-of-thought (CoT) generation.
>
> In this work, we focused on developing STAR-LDM as a base language model (trained via self-supervised pre-training only) and compared it against similarly pre-trained base models to isolate the impact of our architectural innovations. Exploring the adaptation of post-training practices, such as those used to elicit token-space CoT or other explicit reasoning behaviors, to the STAR-LDM framework is an exciting direction for future work. We believe that combining STAR-LDM's latent planning with explicit token-space reasoning strategies could potentially yield even more powerful models, and we will add a brief discussion to Section 9 (Related Work) to highlight these distinctions and future possibilities.
>
> **Re: Related work**
>
> We thank the reviewer for bringing the paper "Controlling Large Language Model with Latent Actions" (Jia et al., 2025) to our attention. This work appeared on arXiv on March 27, 2025, one day before the submission deadline, and we were not aware of it at the time of our initial submission.
>
> Upon review, we agree that it is interesting related work. Their CoLA framework appears to use an inverse dynamics model to extract discrete action tokens, while our STAR-LDM focuses on planning in a continuous sentence embedding space using diffusion. We will cite and discuss this work in the related work section of the final version.

---

> > ### Comment · Reviewer_fPq9 · 2025-06-06
> >
> > Thanks to your response. I have raised my score to 6. Diffusion language models and diffusion for language models are drawing attention and discussion recently. Your work may be of interest to the community.

---

### Official Review · Reviewer_2MBE · 2025-05-12

**Rating:** 6
**Confidence:** 3
**Ethics Flag:** 1

**Summary:**

This paper proposes STAR-LDM, a unified architecture that integrates a latent diffusion “thinking” phase into autoregressive language modeling. The model pauses token generation to perform semantic planning in a continuous embedding space, then resumes generation guided by the refined latent. This design improves coherence, controllability, and performance on language understanding tasks.

**Reasons To Accept:**

Innovative architecture: The integration of diffusion-based semantic planning introduces non-local guidance that complements the myopic nature of standard autoregressive generation, enabling the model to produce globally coherent and semantically consistent text.
Empirical performance: The model achieves strong results on multiple NLU benchmarks, especially commonsense reasoning tasks, outperforming comparable autoregressive baselines.
Theoretical grounding: The use of a variational lower bound (ELBO) provides a principled scoring function for candidate answers in multiple-choice QA, offering theoretical justification for improved accuracy on NLU tasks.

**Reasons To Reject:**

Time complexity analysis is missing: Diffusion models require iterative denoising steps, which, even when applied in latent space, can significantly slow down training and inference compared to standard autoregressive generation.
Planning tied to prefix quality: The quality and effectiveness of the generated latent plan are still inherently dependent on the prefix. If the initial context is ambiguous or incoherent, the planning phase may struggle to provide meaningful long-range guidance.
Lack of explicit control in planning: The latent diffusion-based semantic plan is continuous and implicit. Relying on the latent representation may exacerbate hallucination issues, as any inaccuracy in the denoising process could propagate and mislead the downstream generation.

---

> ### Author Response · Authors · 2025-06-03
>
> We thank Reviewer 2MBE for their valuable feedback and for recognizing the innovative architecture and empirical performance of STAR-LDM. We address the points raised below:
>
> **Re: Time Complexity**
>
> We acknowledge the importance of discussing computational aspects.
> Regarding training complexity, the STAR-LDM training complexity is similar to that of comparable sized LMs becaust training diffusion models does not require iterative denoising steps. For each training sample, we sample an independent noise level t and directly compute the diffusion loss L_DM (Equation 6) based on the noisy embedding z_t. This is done in parallel with the autoregressive language modeling loss L_LM. The addition of the two small DiT modules and the 8 soft prompt tokens used in this work introduces only a minor computational overhead compared to training a standard autoregressive LM of comparable size.
>
> **Inference Complexity**
>
> During inference, STAR-LDM's "thinking" phase involves iterative diffusion steps to refine the semantic plan, which introduces a fixed computational overhead before autoregressive token generation begins. This overhead is then amortized over the tokens generated in the continuation.
>
> We evaluated the inference speed by generating 64-token continuations for 5000 C4 validation examples. The average milliseconds per output token are as follows:
>
> GPT-2 Large (0.77B params): 19.2 ms/token
>
> STAR-LDM (0.96B params): 36.1 ms/token
>
> GPT-2 XL (1.5B params): 35.0 ms/token
>
> As shown, STAR-LDM, which includes 50 diffusion steps for planning in its 768-dimensional latent space, incurs an increase in inference time compared to the similarly-sized GPT-2 Large. Its inference speed (36.1 ms/token) is comparable to the larger GPT-2 XL model (1.5B params vs. 0.96B for STAR-LDM).
>
> Despite GPT-2 XL being approximately 1.5x larger, STAR-LDM consistently outperforms it across all our evaluations, including Natural Language Understanding (Section 5), StoryCloze Generation (Section 6), and C4 Language Generation metrics like MAUVE (Section 7, Table 2). This demonstrates that the computational cost of the latent diffusion planning phase yields substantial improvements in generation quality and understanding capabilities, offering a favorable trade-off when compared to simply scaling up a standard autoregressive model.
>
> We note that diffusion sampling is an active area of research, and adopting methods from recent work on model distillation (e.g., Salimans & Ho, 2022) or one-step diffusion models (e.g., Song et al., 2023) would further optimize the inference time of STAR-LDM in future work. We will add a detailed discussion of these computational considerations to the final version.
>
> **Re: Planning Tied to Prefix Quality**
>
> This is a valid point and a general challenge for prefix-conditioned language models, including standard autoregressive LMs. STAR-LDM's "thinking" phase, which employs an inherently iterative generation process, offers a potential mechanism to mitigate this. By iteratively refining a latent plan, even one initially derived from an ambiguous prefix, the model has an opportunity to converge on a more coherent semantic plan before committing to token generation.
>
> **Re: Explicit Control in Planning and Hallucination Risk**
>
> **Explicit Control**
>
> While the latent plan z is continuous, STAR-LDM offers explicit control over the outcome of this planning phase. As demonstrated in Section 8 (Plug-and-Play Control) and Figure 5, both classifier-free guidance (amplifying the prefix's influence) and classifier-based guidance (steering towards attributes like sentiment or low toxicity) effectively influence the refined plan z_0. Our results (Figures 5, 12, 13, 14) show that this allows for more reliable and fine-grained plug-and-play control over generation attributes compared to standard autoregressive approaches, consistently achieving better fluency-control trade-offs (e.g., reducing positive sentiment from 69% to ~1% while maintaining perplexity).
>
> **Denoising Errors**
>
> Training with noise conditioning (lines 161-167) is key. Perplexity analysis (App. A, Figs. 7-8) shows the AR decoder adapts to plan clarity (z_t). By varying z_t's effective noise, the model interpolates between AR prefix conditioning (at high noise) and strong plan-following (at low noise/clean z_0). This makes the decoder robust to imperfect plans. Adding small noise to z_0 (line 205) and the AR decoder's sequential generation further enhance robustness.
>
> Standard autoregressive models can also suffer from cascading errors during generation. Our StoryCloze evaluation (Section 6, Figure 4) found that STAR-LDM achieved significantly improved narrative coherence and commonsense reasoning compared to baseline autoregressive models of a much larger scale (e.g. Pythia-2.8b). This suggests that, on balance, the semantic planning phase helps to produce more globally consistent and robust generations, rather than exacerbating error propagation.

---

> > ### Comment · Reviewer_UtyY · 2025-06-05
> >
> > Acknowledged, thanks.

---

> > ### Comment · Reviewer_2MBE · 2025-06-11
> >
> > Thanks for your response.

---

### Official Review · Reviewer_UtyY · 2025-05-12

**Rating:** 7
**Confidence:** 3
**Ethics Flag:** 1

**Summary:**

The paper presents a means of incorporating a diffusion model over an embedding space of text into the auto-regressive generation process of a decoder LM. The authors make the claim that this is good based on analogies to how humans produce text. This is plausible but is only speculation. The empirical results however of their method do suggest it does have benefits, albeit compared to baselines it is not always better, but is on average.

**Questions To Authors:**

- What is it about the HS, WG and PIQA tasks that meant the model performed worse than Pytha-1.4b?
- How often is the token produced by the diffusion process different (or significantly less likely) compared to the base LM's next token (or distribution)?

nit: there's an incorrect reference on line 247.

**Reasons To Accept:**

Results are presented on public datasets against a few baselines**. Overall these show that for similar sized models, the proposed approach does improve on the story generation task consistently. It has mixed performance compared to others on the NLU task, although is stronger on average.

**I believe there are other papers which have included some form of diffusion into the discrete text-token generation problem though, but can't cite them from memory as I'm not particularly familiar with the literature. I think it's worth checking this aspect, and what other reviewers comment.

**Reasons To Reject:**

What are the results of the largest/best LM's on these tasks? There is an implied argument here that the "reasoning" of the diffusion process introduced is necessary, however I do wonder what the results of the largest and best LMs are on these benchmarks? I acknowledge it's a difficult comparison due to not controlling for parameter size/model capacity.

The arguments in section 4 that the diffusion produced "semantic plan" is important are weak imho. It's subjective whether these highlighted words are more or less important. What happens with less plausible forced continuations? What happens if the "semantic steering" is rather different to the prompt, does the model follow the prompt or the steering? There are several aspects here which need more analysis.

The proposed model has mixed results in table 1 and 2 compared to the Pythia models.

---

> ### Author Response · Authors · 2025-06-03
>
> We thank Reviewer UtyY for their positive assessment and insightful questions.
>
> **Re: Other diffusion papers**
>
> Our related work (Sec. 9) discusses discrete diffusion models. STAR-LDM differs by employing *latent diffusion planning* via a *continuous sentence embedding* for a "stop-think-autoregress" cycle, a key distinction we'll expand on in the final version.
>
> **Re: Results from larger LLMs on NLU benchmarks**
>
> Our primary goal was to demonstrate the modeling innovation of STAR-LDM. Thus, we focused comparisons on models of similar scale (e.g., GPT-2 XL, Pythia-1.4B), even though these baselines were often trained on significantly more data (e.g., 300B tokens for Pythia vs. 16B for STAR-LDM).
> We agree that contextualizing performance with larger models is valuable. Below, we provide zero-shot NLU average scores for the larger Llama-3.1 8B, which was trained on orders of magnitude more data, alongside our STAR-LDM:
>
>     | Model          | Params | Training Tokens (Approx.) | Avg. NLU |
>     |----------------|--------|---------------------------|----------|
>     | STAR-LDM       | .96B   | 16B                       | 47.1%    |
>     | Pythia-1.4B    | 1.4B   | 300B                      | 46.7%    |
>     | Llama-3.1 8B   | 8B     | 15T+                      | 62.8%    |
>
> This comparison highlights the competitive performance of our ~1B parameter STAR-LDM given its significantly smaller training dataset, while also underscoring the expected performance gains from massive scale and data.
>
> **Re: Influence of semantic plan**
>
> We provide additional analysis on the semantic plan's influence in Appendix A. Figures 7 and 8 show perplexity for teacher-forced and sampled continuations conditioned on embeddings with varying noise levels. These demonstrate:
>
> 1.  Continuations are significantly more probable (lower perplexity) when conditioned on their corresponding clean semantic embedding (`z_0`) compared to conditioning only on the prefix (autoregressive baseline). For instance, in Figure 8 (right), sampled continuations achieve a mean perplexity of ~11 with `z_0` conditioning, versus ~19 for pure autoregressive generation.
>
> 2.  The noise level in `z_t` allows smooth interpolation of the semantic plan's influence at inference time, as shown by the perplexity curves in Figure 7,8.
>
> The strong adherence to the semantic plan is also evident in our sentiment steering results (Figure 5a, Appendix D Figure 12a), where classifier guidance on the continuous embeddings effectively controls output sentiment while maintaining fluency.
>
> We agree that our approach opens many avenues for analysis that we are excited to explore in future work.
>
> **Re: NLU Performance variations**
>
> STAR-LDM combines two core components: **latent diffusion planning** for global semantic coherence and **autoregressive language modeling** for local linguistic precision. Our NLU results in Table 1 reflect their complementary strengths:
>
> 1.  **Diffusion for Global Semantics:** The diffusion component (scoring via "Diffusion Loss") excels when tasks require assessing the conceptual alignment of an answer. This is evident on benchmarks like CSQA, where its standalone performance (46.9) is strong, indicating it effectively captures high-level semantic plausibility. Benchmarks like CSQA, SIQA, OBQA, and ARC benefit significantly from the diffusion planner's ability to ensure answers are semantically on-target globally.
> 2.  **Autoregression for Local Nuance:** The autoregressive component (scoring via "LM Loss") focuses on token-level predictions and fine-grained linguistic details. HS and WG often hinge on subtle word choices or precise linguistic links (e.g., pronoun resolution in WinoGrande based on quantifiers like 'less'/'more' time) that a sentence-level semantic plan might not fully capture. Here, the AR model's local reasoning is more critical.  PIQA's physical commonsense might be less well-represented in the general sentence embeddings used for diffusion planning compared to what an AR model might implicitly learn from vast text.
>
> The current balance may favor tasks emphasizing broad coherence. Future work could further explore adjusting the influence of these components.
>
> **Re: How often the diffusion process leads to different/less likely tokens vs. base LM**
>
> Our perplexity analyses in Appendix A (Figures 7 & 8) address this. Conditioning on the semantic embedding (`z_0`) makes the sampled continuation significantly more likely (e.g., perplexity ~11 vs. ~19 for sampled continuations). This indicates a substantial shift in token probability distributions. The autoregressive LM alone must model the full distribution of all continuations, whereas the semantic plan (via `z_0`) effectively "commits" the model to a specific high-level meaning, narrowing the probable token sequences.

---

### Official Review · Reviewer_iRQd · 2025-05-14

**Rating:** 1
**Confidence:** 5
**Ethics Flag:** 1

**Summary:**

This work violates the double-blind policy. L68 says "our work Lovelace et al. (2024)"

**Reasons To Accept:**

N/A

**Reasons To Reject:**

N/A

---

> ### Author Response · Authors · 2025-05-28
> **Important Clarification**
>
> We acknowledge that the sentence construction creates two possible interpretations, and we understand the reviewer's concern. However, **to clarify our intent: "our work" refers to the current submission, *not* to Lovelace et al. (2024).** This interpretation is consistent with our usage of "our work" throughout the paper (lines 100, 320, and 333), where it always refers to the present submission.
>
> In hindsight, we should have restructured the sentence when adding the citation—perhaps "the sentence embeddings we adopt \cite{lovelace2024}" would have been clearer. We sincerely apologize for this mistake and regret this ambiguity. We appreciate the reviewer bringing it to our attention and will fix it in the final version.

---

> > ### Comment · Reviewer_iRQd · 2025-06-07
> >
> > Thanks for responding. There was no ambiguity about the meaning of "our work" at all the other places. The one instance I mentioned does interpret to mean your work = Lovelace et al. I do understand that no author will knowingly break the double-blind policy, but this mistake was indeed made (even if unintentionally), so I won't review this work. I am sorry.

---

### Decision · Program_Chairs · 2025-07-08

**Decision:**

Accept

**Comment:**

STAR-LDM proposes a novel integration of a latent diffusion “thinking” phase into standard autoregressive text generation, pausing token production to perform semantic planning in a continuous embedding space before resuming guided decoding.

Empirically, it achieves consistent gains over same-sized autoregressive baselines on story generation tasks and shows mixed but generally positive improvements on commonsense and multiple-choice QA benchmarks, supported by a principled ELBO‐based scoring framework.

However, it also has notable shortcomings. As highlighted by R2, the human-analogy motivation for the diffusion phase remains speculative and the presented “semantic plans” are subjective, with no robustness analysis under divergent or adversarial steering. R3 emphasizes the absence of any detailed time-complexity or latency evaluation—iterative denoising in latent space could significantly slow both training and inference. Finally, R4 points out that the model’s reliance on external Sentence-T5 embeddings is never scrutinized, leaving open questions about how embedding quality or choice affects planning effectiveness and downstream generation.

During rebuttal, R4 raised the score and acknowledged the efforts of the authors. Overall, I think this paper presents interesting idea which would be very useful for the community to combine diffusion-style thinking in LLM generation.

[Automatically added comment] At least one review was discounted during the decision process due to quality]